# Mucosal Associated Invariant T Cells in Cancer-Friend or Foe?

**DOI:** 10.3390/cancers13071582

**Published:** 2021-03-30

**Authors:** Chloe O’Neill, Féaron C. Cassidy, Donal O’Shea, Andrew E. Hogan

**Affiliations:** 1Lonsdale Human Health Institute, Maynooth University, W23 F2H6 Maynooth, Ireland; Chloe.ONeill@mu.ie (C.O.); Fearon.Cassidy@mu.ie (F.C.C.); 2St Vincent’s University Hospital, University College Dublin, D04 T6F4 Dublin, Ireland; info@dosheaendo.ie; 3National Children’s Research Centre, D12 HX96 Dublin, Ireland

**Keywords:** MAIT cells, cancer, immune checkpoint, metabolism

## Abstract

**Simple Summary:**

Mucosal associated invariant T (MAIT) cells are a population of innate T cells which play an important role in host protection against bacterial, fungi and viruses. MAIT cells are armed with potent cytotoxic machinery, allowing them to kill invading pathogens but also transformed cells. Many studies have investigated MAIT cells in the setting of various human cancers, with conflicting results. Several studies have identified MAIT cells as potent anti-cancer effector cells, whilst others have reported a potential pathogenic role for MAIT cells in certain cancers. In this review, we discuss the current knowledge base and highlight potential mechanisms which may underpin the reported cancer-related alterations in MAIT cells.

**Abstract:**

Mucosal associated invariant T (MAIT) cells are a population of unconventional T cells which can bridge the innate and adaptive immune systems. Well-described roles for MAIT cells include host protection against invading bacteria, fungi and viruses. Upon activation, MAIT cells become prolific effector cells, capable of producing a range of cytokines and lytic molecules. In addition to their anti-microbial role, MAIT cells have been implicated in immune responses to cancer, with opposing beneficial and pathogenic roles reported. On the one hand, MAIT cells can home to the site of the tumour in many human cancers and can produce anti-tumour molecules. On the other, MAIT cells can display defective phenotypes in certain cancers and produce pro-tumour molecules. In this review, we discuss the current literature on the diverse roles for MAIT cells in cancer, outlining their frequencies, functions and associations with N staging and prognosis. We also discuss potential mechanisms underpinning cancer-related alterations in MAIT cells and highlight therapeutic approaches to harness or target MAIT cells in cancer.

## 1. Introduction

Mucosal Associated Invariant T (MAIT) cells are a population of unconventional T cells which are enriched in mucosal tissues such as the lung and gut but are also present in other tissues including the skin, adipose tissue and the liver [1,2,3,4]. In humans, MAIT cells represent between 1–8% of T lymphocytes in the blood, up to 10% of intestinal T cells and up to 50% of all T cells in the liver [2]. Unlike conventional T cells, MAIT cells are not restricted by MHC but recognize the MHC-related protein MR1. MAIT cells express a semi-invariant TCR α-chain (Vα7.2-Jα33/20/12 in humans) and a limited repository of TCR-β chains, mostly from the TRBV20 and TRBV6 gene families [5,6]. They recognize vitamin B metabolites synthesized by bacteria and yeast that are bound to MR1 molecules [7,8]. Vitamin B metabolism pathways are conserved among a broad range of species, enabling MAIT cell recognition of a diverse range of pathogens [9,10]. MAIT cells can also be activated independent of TCR engagement, via pro-inflammatory cytokines such as IL-18 [11,12,13]. Upon activation, MAIT cells can rapidly respond, producing a milieu of cytokines including interferon-gamma (IFNγ), tumour necrosis factor alpha (TNFα) and interleukin-17 (IL-17) [12,14]. MAIT cell cytokine profiles can vary depending on their tissue localization [15]. Peripheral MAIT cells are potent producers of IFNγ and TNFα, whereas IL-17 producing MAIT cells are rare in the periphery but are more abundant in, for example, the female genital mucosa [16]. We have also noted IL-10 production by MAIT cells in the adipose tissue but not the periphery of healthy individuals [3]. In addition to cytokine production, MAIT cells are also prolific producers of cytotoxic proteins such as perforin, granzymes and granulysin (Figure 1) [9,12,17]. Granule exocytosis is the most common mechanism by which cytotoxic cells mediate apoptosis in tumour cells, relying on granzymes to enter the tumour cell cytoplasm via pores formed by perforins [18]. MAIT cells express granzymes A and K constitutively and upon activation upregulate perforin and granzyme B expression [17]. Armed with this killing machinery, MAIT cells have been shown to be potent killers of transformed cells [19,20]. When paired with their reported presence within numerous human tumour types, these findings have brought MAIT cells into focus in the setting of cancer, highlighting them as a potential novel target for cancer immunotherapy. In this review, we outline the current literature on MAIT cells in cancer, discussing their opposing roles, mechanisms of dysregulation and potential as a therapeutic agent.

## 2. MAIT Cells in Cancer

MAIT cells have been detected within numerous human tumour types, such as cervical, colorectal, oesophageal, liver, kidney and brain cancer as well as having modified characteristics in the blood of people with multiple myeloma (Table 1) [19,21,22,23,24,25,26,27,28]. The impact of the cancer on MAIT cell frequency, phenotype and function, and conversely the impact of MAIT cells on the cancer can vary greatly from cancer to cancer. We will now discuss MAIT cells in individual cancer types.

### 2.1. Colorectal Cancer

Colorectal cancer (CRC) typically develops from the growth of adenomatous polyps which can lead to impairments in epithelial barrier function and disrupted mucosal homeostasis, resulting in inflammation [29]. The risk of CRC is also mediated by diseases such as ulcerative colitis and Crohn’s disease. Several studies have investigated MAIT cells in the setting of CRC and have found that MAIT cells are enriched in neoplasms and CRC tumours compared to healthy colon or peripheral blood [25,30]. Li et al., also investigated MAIT cell infiltration in CRC tumours, and determined that infiltration was tumour site-specific, as MAIT cells were enriched in CRC when compared to non-small cell lung cancer (NSCLC) and renal cell carcinoma (RCC) tumours [31]. Increased MAIT cell infiltration into tumours was associated with reduced overall survival (OS) and disease-free survival (DFS) in patients living with CRC, potentially suggesting a pathogenic role within colorectal cancer [32]. Functionally MAIT cells were found to be impaired, with reduced IFNγ production [21,22]. This impairment was associated with proximity to the tumour, with tumour-margin MAIT cells producing more IFNγ than tumour-infiltrating MAIT cells [22]. In addition to reduced IFNγ production, Kelly and colleagues demonstrated that chronically activated MAIT cells are potent producers of IL-13. The receptor for IL-13 is abundantly expressed in CRC tumours and precancerous polyps, and can act as a pro-tumour factor [33]. Reduced IFNγ production and the production of IL-13 may explain the negative association with MAIT cells and outcomes in CRC. However, other studies have demonstrated that MAIT cells retain their IFN-γ production and display a strong cytotoxic function within CRC tumours [25,30]. In addition to IFNγ, Ling and colleagues also reported elevated IL-17A production by MAIT cells in both the periphery and tumours of patients with CRC [30]. IL-17 producing MAIT cells have been implicated in numerous chronic inflammatory diseases [14], with one study demonstrating a negative role for IL-17 producing MAIT cells in murine models of melanoma, driven by IL-17 inhibition of natural killer (NK) cell responses [34]. Whether MAIT cells have a positive or negative role in anti-tumour immunity in CRC remains a point of debate and will require further work to elucidate the mechanisms underpinning the reported alterations.

**Table 1 cancers-13-01582-t001:** Overview of MAIT cell frequencies, cytokine profile and cytotoxic molecule production in cancer.

Cancer Type	MAIT Cell Frequencies	MAIT Cell Cytokines	MAIT Cell Cytotoxic Molecules	Ref
Colorectal	↑ in tumour	↓ IFNγ	↑Granzyme B	[15]
	↑ in tumour		↑ Granzyme B ↑ Perforin ↑ CD107a	[25]
	↓ in periphery ↑ in tumour	No change IFNγ, IL-17, TNF-α	↑Granzyme B ↑ Perforin	[20]
	↓in CRC liver metastases	↓ IFNγ	↓ Granzyme B	[22]
	↓ in periphery ↑ in tumour	↓ IFNγ in periphery ↑ IFNγ in tumour ↑ IL-17 in tumour		[30]
	↑ in tumour			[31]
	No change	↑IL-13		[33]
Multiple Myeloma	↓ in periphery	↓ IFNγ, IL-17, IL-22 in untreated patients		[19]
Breast	No change in periphery	↑ IL-17 in breast duct ↓ IFNγ in breast duct		[28]
Hepatocellular	↓ in periphery & tumour	↓ IFNγ, IL-17, IL-8	↓ Granzyme B ↓ Perforin	[26]
Oesophageal	↓ in periphery ↑ in tumours	↓ IFNγ & TNF-α		[23]
Gastric	↓ in periphery	Unchanged IFNγ & TNF-α ↓ IL-17	↓ Granzyme B	[35]
Lung	↓ in periphery ↑ in cancer tissue	Unchanged IFNγ, TNF-α, IL-17	↑Granzyme B ↑ Perforin	[20]
Kidney		IFNγ & IL-17		[27]
Cervical	↓ in periphery ↑ in periphery			[36] [24]
Melanoma	↓ in periphery of non-responders to PD-1 therapy	↑ IFNγ in periphery of non-responders to PD-1 therapy	↑ Granzyme B in periphery of non-responders to PD-1 therapy	[37]

### 2.2. Multiple Myeloma

Multiple Myeloma (MM) is a haematological malignancy that is distinguished by the uncontrolled outgrowth of plasma cells from the bone marrow [38]. In 2018, Gherardin et al. reported reduced MAIT cell frequencies in patients with untreated MM compared to healthy controls. The decline in frequency was even greater in patients with refractory/relapsed (R/R) MM [19]. The reduced frequencies were not associated with redistribution to the bone marrow, but elevated expression of CD27 was noted suggesting an exhausted phenotype. Functionally, MAIT cells from untreated MM expressed lower levels of IFNγ production, but this was restored in R/R MM patients. The exact role MAIT cells play in MM remains unclear, however the authors did demonstrate that MAIT cells were capable of killing myeloma cell lines suggesting potential for harnessing MAIT cells as an immunotherapy [19].

### 2.3. Breast Cancer

Breast cancer is the most frequent malignancy in women globally [39]. Although MAIT cells have not been investigated in primary breast tumours, Zumwalde and colleagues showed that healthy human breast ducts contain a population of MAIT cells bearing a distinct T cell receptor Vβ usage [28]. Breast ducts contain higher frequencies of IL-17 producing MAIT cells in comparison to the periphery. In the same study, the authors showed that MDA-MB-231 breast carcinoma cells were capable of activating breast MAIT cells in a TCR-ve, MR1-ve, and microbe-dependent manner, resulting in a strong IL-17 bias [28]. Again, the role for IL-17 in cancer remains unclear, with early IL-17 production supporting tumour growth but offering a protective role in established cancer [40]. The negative impact of IL-17 producing MAIT cells on cancer progression might suggest MAIT cells are early producers [34], highlighting a novel therapeutic avenue in cancers such as breast cancer, however further work on the observations by Zumwalde and colleague is required to elucidate a role for MAIT cells in primary breast cancer.

### 2.4. Hepatocellular Carcinoma

Liver cancer is the second leading cause of cancer-related deaths globally. Hepatocellular carcinoma (HCC) represents nearly 90% of all cases of primary liver cancer [41]. The liver is home to the highest frequency of MAIT cells in humans, up to 50% of intrahepatic T cells [2,42]. Duan and colleagues reported an association between activated/exhausted MAIT cells and poor prognosis in HCC [26]. MAIT cells from HCC tumours expressed higher levels of the inhibitory markers PD-1, CTLA-4 and TIM-3 when compared to healthy liver [26,32]. Frequencies of MAIT cells were lower in both the periphery and tumours of patients with HCC, with the authors suggesting the reduction in tumour infiltrating MAIT cells was a reflection of liver cirrhosis opposed to increased apoptosis [26]. HCC tumours typically develop in a fibrotic or cirrhotic liver [41]. MAIT cells have been shown to be a profibrotic population in chronic liver disease [43]. Using an in vitro system, Hedge and colleagues demonstrated that MAIT cells could drive a pro-inflammatory human hepatic myofibroblast phenotype in an IL-17/TNFα dependent manner [43]. This suggests that MAIT cells could contribute to the progression of HCC. However, MAIT cells from HCC tumours showed reduced IFNγ and IL-17 production but did upregulate the pro angiogenic cytokine IL-8, again suggesting a pro-tumour role [26]. Further investigations are required to establish if MAIT cells represent a pathogenic pro-tumour population in HCC, and must consider co-morbid viral infection (a major risk factor for the development of HCC) as a contributor to the observed phenotype [32].

### 2.5. Oesophageal Cancer

Oesophageal cancer is the sixth leading cause of cancer death in the world [44]. Two major subtypes exist, oesophageal squamous cell carcinoma (OSCC) and adenocarcinoma (OAC) which are epidemiologically and pathologically distinct. OAC typically develops from Barrett’s mucosa in the lower oesophagus and has a poor prognosis. Barrett’s oesophagus (BO) is the name used for its pre-malignant inflammatory condition. Melo and colleagues found that the risk of death with OAC is increased with low levels of tumour-infiltrating MAIT cells [23]. This is in agreement with Yao et al. who reported improved prognosis with higher MAIT cell infiltration in OAC [32]. Healthy MAIT cells were able to kill OAC cell lines in vitro, interestingly this was not impacted by the addition of OAC tissue conditioned media, though MAIT cell production of IFNγ and TNFα was diminished [23]. The impact of the tumour microenvironment (TME) has emerged as a major research area in immunotherapy but remains relatively unexplored with respect to MAIT cells. This will be discussed in detail in Section 2.13.

### 2.6. Gastric Cancer

Gastric cancers, with gastric adenocarcinoma (GAC) as the most common histological type, represents a considerable global health burden [45]. A recent study by Shao et al. reported that MAIT cells in the peripheral blood were significantly lower in patients with gastric cancer in comparison to healthy controls. Functionally, MAIT cells from patients with gastric cancer produced comparable levels of IFNγ and TNFα when compared to healthy controls, demonstrating a preservation of Th1 cytokines, and produced almost undetectable levels of IL-17 [35]. MAIT cell frequencies and function within the tumour were not investigated and will require future studies to determine if MAIT cells are associated with better or worse prognosis in gastric cancer.

### 2.7. Lung Cancer

Lung cancer is one of the most frequently diagnosed cancers and is the leading cause of cancer-related death worldwide. Non-small-cell lung cancer (NSCLC), a heterogeneous class of tumours, represents approximately 85% of all new lung cancer diagnoses [46]. Small-cell lung cancer (SCLC) makes up the remaining 15% and is marked by an exceptionally high proliferative rate, early metastasis and poor prognosis [47]. Won and colleagues have demonstrated reduced MAIT cell frequencies in the periphery of patients with lung cancer. Reduced absolute MAIT cell numbers were associated with N staging, suggesting MAIT cells may imply the stage of cancer progression, however this was in a cohort of mixed cancers, not lung cancer specifically and thus requires further investigation. A study by Yan et al., investigated the role of MAIT cells in a murine model of lung metastasis, and demonstrated reduced tumour initiation, development and metastasis in MAIT cell deficient mice [34]. The authors elegantly demonstrated that MAIT cells suppress NK cell anti-tumour effector functions in a MR1 and IL-17 dependent manner, providing mechanistic evidence for a pathogenic role for MAIT cells [34]. Translational investigations in humans are required to establish if MAIT cells are a true foe in lung cancer, and if they may represent a novel therapeutic target.

### 2.8. Kidney Cancer

Kidney cancer develops from the renal parenchyma with roughly 70% of cases being renal cell carcinomas [48]. To date, little research has investigated the presence and role of MAIT cells in kidney cancer. In 2008, Peterfalvi et al. detected the invariant Vα7.2-Jα33 TCR that is characteristic of MAIT cells in kidney tumours, suggesting an infiltration of MAIT cells. Kidney tumours with Vα7.2-Jα33 TCRα expression were infiltrated by CD8+ T cells with HLA-DR expression, suggesting an activated state. Local activation of MAIT cells within tumours was suggested to be possible due to the co-expression of MR1 with the invariant MAIT TCR in 10 out of 11 kidney tumours [27].

### 2.9. Cervical Cancer

Cervical cancer develops from cervical intraepithelial neoplasia and often as a result of persistent human papilloma virus (HPV) infection [49]. Lu et al. found that CD8+, CD4+, and highly activated CD38+CD8+MAIT cells were significantly increased in the periphery of cervical cancer patients in comparison to healthy donors [24]. This alteration positively correlated with monocytic myeloid-derived suppressor cells (Mo-MDSC). CD8+ MAIT cells displayed higher expression levels of the activation marker CD38 in cervical cancer patients versus healthy donors. Double negative (DN) and PD1+ DN MAIT cells were reduced in the peripheral blood, and a positive association was found between elevated circulating PD1+ DN MAIT cells and progression free survival [24]. Huang and colleagues also investigated the role of MAIT cells in cervical cancer. In contradiction with the study by Lu and colleagues, they found that the percentage of circulating MAIT cells was significantly lower in cervical cancer patients in comparison to healthy controls, even after age adjustment [36]. Huang et al. recruited individuals without pre-malignant disease or any gynaecologic malignancy as healthy controls, whereas Lu et al. enrolled patients with uterine leiomyoma (uterine fibroids) as controls. It may therefore be possible that the contrasting findings regarding MAIT cell frequency may be contributed to by the presence of benign tumours in one group of controls, again more detailed analysis is required to determine the exact role for MAIT cells in cervical cancer.

### 2.10. Melanoma

Melanoma is a malignancy of melanocytes, pigment-producing cells which are found in the skin, iris and rectum [50]. The cutaneous form is responsible for over 55,000 deaths annually [51]. A study by De Biasi and colleagues investigated T cell populations in patients with melanoma commencing anti-PD-1 therapy, and found that a MAIT cell frequency greater than 1.7% of peripheral T cells at baseline predicted responsiveness to anti-PD-1 therapy [37]. The authors also noted that MAIT cells with an activated (CD69) and homing (CXCR4) phenotype were more abundant in melanoma patients responding to anti-PD-1 therapy [37]. Cytokine production was also different in responders versus non-responders with higher production of IFNγ and granzyme B, perhaps suggesting a protective and beneficial role in melanoma.

### 2.11. Mechanisms of Dysregulation

As discussed, MAIT cells display a wide range of alterations in cancer, ranging from reduced frequencies in the periphery to altered cytokine profiles. Whilst several of the studies reviewed have highlighted a potential role for MAIT cells in cancer as a prognostic marker or therapeutic target, few studies have provided mechanistic evidence underpinning the observed alterations. Drawing on the data available and evidence from other cell populations, we will discuss some potential mechanisms of MAIT cell dysregulation in cancer (Figure 2).

### 2.12. Immune Checkpoints

Immune checkpoints refer to a range of inhibitory pathways hardwired into the immune system. These are critical for maintaining self-tolerance and regulating the immune response in order to minimize collateral tissue damage [52]. It is now well established that tumours can target this pathway for immune evasion, in particular avoiding T cell responses. Two of the most prominent immune checkpoints are PD-1 and CTLA-4. Engagement of these molecules with their respective ligands results in inhibition of T cell responses [52]. Several studies have highlighted the expression of these immune checkpoints on MAIT cells in human cancers [23,26,31]. Targeting of these immune checkpoints has been highlighted as one of the biggest breakthroughs in cancer therapeutics [53]. Another checkpoint receptor emerging as a potential cancer therapeutic target is T cell immunoglobulin and mucin domain-containing protein 3 (TIM-3) [54]. TIM-3 is highly expressed on dysfunctional and exhausted T cells and can limit IFNγ production by conventional T cells. TIM-3 expression was noted on MAIT cells in CRC and HCC [26,31]. To date, studies investigating the impact of checkpoint inhibitors on MAIT cell functions are limited but targeting MAIT cells with checkpoint inhibitors may help harness their anti-tumour effector functions such as IFNγ and cytolytic molecule production.

### 2.13. Tumour Microenvironment

Another intense area of research in cancer is the immunosuppressive landscape of the TME. The TME is a highly complex and dynamic ecosystem controlled by the tumour to support its growth and survival [55]. The TME drives immunosuppression via altered levels of metabolites, cytokines, nutrients and oxygen [55]. The TME is also associated with elevated expression of immune checkpoints. Not all patients with cancer respond to checkpoint monotherapy, this is proposed to be due to differences in the TME [56]. Although the impact of the TME has not been directly investigated in MAIT cells, many studies are emerging in other cell types which provide possible avenues for further exploration. One area of great interest is immunometabolism. The harsh TME forces infiltrating cells to undergo metabolic adaptations and can undermine the effectiveness of their responses [57]. Cancer cells are highly glycolytic, this results in limited availability of glucose which can impact the effector functions of infiltrating immune cells [58]. Our research group and others have recently outlined the metabolic requirements of MAIT cells [59,60,61]. MAIT cells depend on glycolysis for their production of IFNγ, demonstrating reduced production under acute glucose restriction [60]. Whether the reduced IFNγ production by MAIT cells in cancer is linked to restricted glucose availability requires further investigation. Furthermore, glucose restriction can lead to increased rates of apoptosis in conventional T cells [62,63], which might explain the reduced frequencies of MAIT cells in certain cancers [19]. Increased rates of aerobic glycolysis result in elevated levels of lactate which acidifies the TME and can limit immunosurveillance, and T and NK cells [64,65,66]. Although not investigated to date, another potential driver of MAIT cell dysfunction within the TME is hypoxia, as tumour mass increases it can become hypoxic which can directly inhibit T cell function [54,67,68]. Another unknown is the signals driving the activation of MAIT cells in cancer. Tumour antigen specific T cells have been reported in numerous cancers, however, to date, tumour-derived antigens capable of activating MAIT cells have not been identified [69,70,71]. It is well established that the TME contains a milieu of inflammatory cytokines including IL-18 [72], a known driver of MAIT cell activation [11,13]. Interestingly IL-18 has been demonstrated to drive PD-1 dependent immunosuppression in cancer [73]. Whether IL-18 in the TME drives the activation of MAIT cells and/or the expression of PD-1 remains to be explored.

## 3. Conclusions

As highlighted, MAIT cells have diverse roles in cancer. On the one hand, MAIT cells can home to the tumour site and elicit Th1-cytokine response and upregulate cytolytic granule expression [20,25,30]. Conversely, MAIT cells can display an exhausted phenotype with reduced capacity to produce anti-tumour cytokines, such as IFNγ and TNFα [21,22,23]. Due to their potent effector functions and abundant frequencies in humans, MAIT cells may represent a very attractive therapeutic target for cancer. This is supported by the identification of a MR1-restricted T cell clone capable of killing multiple types of cancer in a TCR-MR1 dependent manner [74]. MAIT cells with their increased expression of PD-1, CTLA-4 and TIM-3 in cancer may be targets for immune-checkpoint inhibitor therapy, potentially restoring MAIT cells anti-tumour abilities. This is supported by the observation that MAIT cell frequency predicted responsiveness to PD-1 therapy [37]. Recently Parrot et al. demonstrated that human MAIT cells could be expanded and engineered for immunotherapy resulting in enhanced cytolytic capacity [75]. However, further research is urgently required to establish if MAIT cells are a friend or foe in cancer, along with the mechanisms underpinning their alterations.

## Figures and Tables

**Figure 1 cancers-13-01582-f001:**
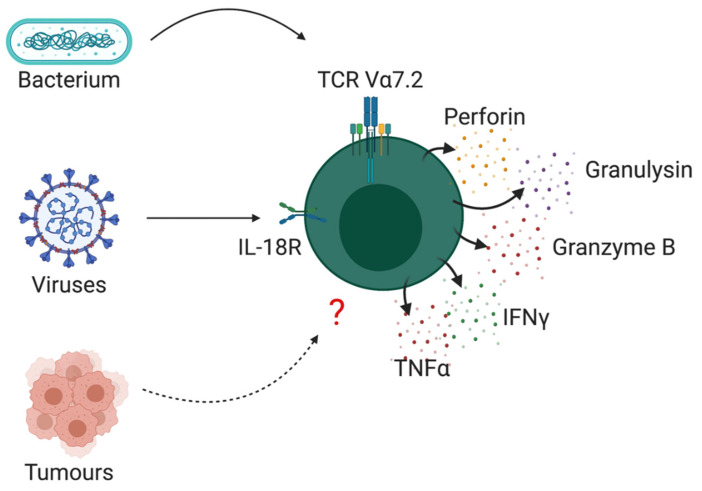
Schematic outlining MAIT cells effector cytokines and killing machinery.

**Figure 2 cancers-13-01582-f002:**
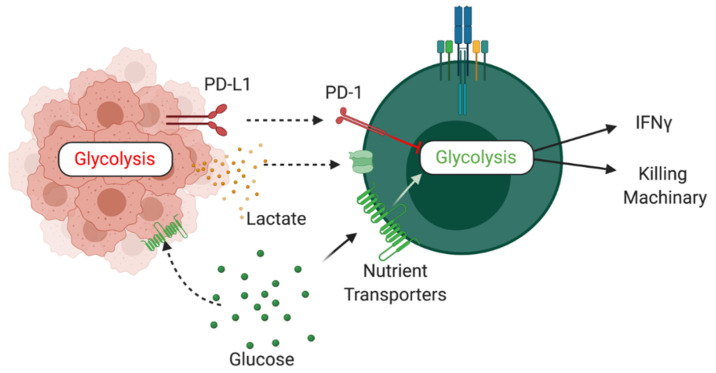
Schematic outlining potential mechanisms of cancer induced dysregulation of MAIT cells.

## Data Availability

Not applicable.

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
