# Peer review of "Mucosal Associated Invariant T Cells in Cancer-Friend or Foe?"

_cancers, 2021, doi:10.3390/cancers13071582_

Round 1

Reviewer 1 Report

This review on MAIT cells in cancer is very well written and comprehensive. It's well structured, has appropriate detail and the figures and tables are helpful additions to the manuscript.

There are a couple of typos/grammatical throughout (eg Page 1, line 20; page 2 line 50; page 3 line 80/81). This is not a big deal, but a careful read through the manuscript is recommended.

The authors may wish to include the recently published De Biasi et al, Nature Comms, (2021) 2:1669, as this paper, although very recent, seems pertinent to this review.

A final area worth considering some commentary on would be the few papers on non-MAIT MR1-restricted T cells (lepore et al, Elife and Crowther et al, Nature Immunology) which appear to have MR1-dependent anti-tumour properties. This is perhaps not necessary for this review which focusses on TRAV1-2+ MAIT cells, but this would nonetheless be a worthy addition.

Author Response

The authors would like to thank the reviewer for their time & expertise in reviewing our manuscript. We were very pleased that they found it "very well written and comprehensive". We have addressed their points (outlined below) and feel it has strengthened our review.

We note that there was several typos/errors and we have carefully reviewed our manuscript correcting these - the authors would like apologise for these.

As suggested by the reviewer we have added the recent study on MAIT Cells in Malignant Melanoma by De Biasi et al.

We have also briefly discussed the published works on MR1+ T cells as suggested by the reviewer. 

Reviewer 2 Report

In this review, O’Neill et al give a comprehensive overview of the studies performed to date focusing on MAIT cell presence and effector functions in different types of tumors. The authors concentrate on studies on human subjects, which is probably most relevant, as mouse and human MAIT cell biology appears to differ substantially. The manuscript is well written and easy to follow, and I only have a few comments.

It would probably be helpful to the reader to mention that the cytokine profile in MAIT cells appear to differ depending on the organ where they reside. For example, intestinal, hepatic and blood-derived MAIT cells have a strong IFN-gamma bias, while MAIT cells from breast ducts and female genital tract are more prone to IL-17 production. As the balance between Th1 and Th17 type responses may be one important factor to determine the efficiency of the local immune response in tumors, the baseline cytokine production by MAIT cells in different organs would be interesting as background information when discussing different types of tumors. See for example Dias et al (2018, Front Immunol 9:1602) for an introduction to MAIT cell cytokines in different organs.

It would be useful if the authors could speculate a bit more on potential MAIT cell activation modes (antigen, cytokines, TLR ligands… ?) in the TME.

There are a few typos that make some of the sentences hard to follow, i.e. lines 80, 128-129, 153.

Author Response

The authors would like to thank the reviewer for their time and expertise in reviewing our manuscript. We are enthused that they found it "well written and easy to follow". We have addressed the comments from the reviewer (outlined below) and feel it has improved our review. 

Firstly, we would like to apologies for the typos & grammatical errors, we have careful reviewed the manuscript to correct these.

As suggested by the reviewer we have included some information on tissue-specific cytokine responses in our introduction and agree that it improves context and accessibility.

We have also discussed potential drivers of MAIT cell activation in the TME as suggested by the reviewer and feel it has strengthened that particular section. 

Reviewer 3 Report

This paper entitled "Mucosal associated invariant T cells in cancer-friend or foe" is an excellent review paper which provides for important information for those who know either little about these invariant T cells, or, for readers wishing to be up-to-date on current research in this area.  Coverage of the literature regarding mucosal associated invariant T cells is both thorough and excellent providing the opposing roles of these T cells in cancer, dysregulation mechanisms, and more importantly, the potential of these invariant T cells to be used therapeutically. Table 1 is exceptionally well noted by this reviewer for its simplicity in providing an overview of these mucosal associated invariant T cells as they are associated with a variety of cancer types, with description of their cell frequencies, cytokines, and cytotoxic molecules. Their association with providing either a cytokine response and up regulates I told lytic granule expression, or, the exhaustive in their function reflective of their reduced capacity to produce such cytokines and other molecules. The thought that these mucosal associated invariant T cells could be engineered for immunotherapy possibilities is important. 

Author Response

We would like to thank the reviewer for their time and very complementary review of our manuscript. We are very pleased with their summary "excellent review paper which provides for important information for those who know either little about these invariant T cells, or, for readers wishing to be up-to-date on current research in this area".

We also appreciate the comment re our table -  "Table 1 is exceptionally well noted by this reviewer for its simplicity in providing an overview"